# Contradictions and Promise for End-of-Life Communication among Family and Friends: Death over Dinner Conversations

**DOI:** 10.3390/bs7020024

**Published:** 2017-04-20

**Authors:** Andrea Lambert South, Jessica Elton

**Affiliations:** 1Department of Communication, Northern Kentucky University, Highland Heights, KY 41099, USA; 2Department of Communication, Eastern Michigan University, Ypsilanti, MI 48197, USA; jelton@emich.edu

**Keywords:** end-of-life, family, friends, communication, death, dying

## Abstract

The free, open-access website called “Let’s Get Together and Talk about Death”, or Death over Dinner (DoD), provides resources for initiating end-of-life conversations with family and friends by taking the frightening—talking about death—and transforming it into the familiar—a conversation over dinner. This qualitative, descriptive study uses grounded theory and thematic analysis to answer the following research question: How do friend and family groups communicate about death and dying in DoD conversations? To answer this question, 52 dinner groups were recruited and conversations were conducted, which consisted of a facilitator and volunteers. The facilitators were the researchers or research assistants who allowed dinner participants to control the conversation and identify topics of interest, and participants were free to share as much or as little as they wanted. Our analysis revealed that family and friend groups communicated similarly in that they talked about similar topics and used similar communication strategies to discuss those topics. Three major themes emerged: *Desire for a good death*, which juxtaposed people’s perceptions of a “dreaded” death with those of a “desirable” death; *tactics for coping*, which consisted of the subthemes of humour to diffuse tension or deflect discomfort, spiritual reassurance, and topic avoidance; and *topics that elicit fear or uncertainty*, which consisted of the subthemes of organ and whole-body donation, hospice and palliative care, wills and advance directives. Ultimately, however, participants felt their experiences were positive and DoD shows promise as a tool for families to engage in end-of-life conversations.

## 1. Introduction

Death is an important stage in the life cycle and, like birth, is an inevitability for all human beings [1]; however, in many cultures death and dying are stigmatised, taboo, or fear-inducing topics [2,3,4,5,6]. This makes communicating about death challenging [7,8], and there are several negative implications of avoiding talking about death. Countries that are the most death-averse, and therefore the least likely to communicate about the end of life, tend to rank lowest in end-of-life care quality [6]. In many countries, the majority of people report that they would like to die at home; however, most people die in hospitals [5,9,10], and the Australian Medical Association reports that most Australians want palliative care, yet few actually receive it [11]. Failure to communicate about end-of-life preferences has been identified as one of the reasons people do not receive the care they prefer [6]. Thus, avoiding end-of-life communication results in greater health care spending, more unwanted hospital admissions, and less patient and family satisfaction [9,12].

Although avoiding communication about death has negative repercussions, engaging in it has many benefits. For example, talking about death may help people work through their fears and better understand what they want during the end of life and also makes one’s care preferences known to others [13]. It might also make people aware of end-of-life services, like palliative care and hospice, of which they previously had little to no knowledge [6], and sharing positive stories about end of life may change people’s attitudes toward death and dying, thus making it easier for people to prepare for the end of life. Communication about the end of life also results in better care for the patient and offers stress relief and support for families and friends [5,14].

Noting the importance of communicating about end-of-life issues, many countries, from Hong Kong to Hungary, are working to destigmatise death and encourage people to discuss and plan for it [12,15,16,17]. Organisations like the National Health Service in the United Kingdom, and the Institute of Medicine in the United States, encourage people to speak with their families and care providers about their end-of-life wishes in order to normalise these conversations [18,19].

## 2. Death over Dinner

In response to health professionals’ and policy makers’ calls for people to communicate more about death and dying, a number of resources have been created that seek to help people engage with others in these conversations. In addition to both community- and web-based resources that encourage conversations about death, such as Death Café, the Conversation Project, and the Before I Die Festivals, is “Let’s Get Together and Talk about Death” (e.g., Death over Dinner or DoD), a free, public website created by Michael Hebb to facilitate discussions of death and dying with family and friends [20]. DoD strives to create a space where people can consider and share their thoughts about death and their preferences for end-of-life care by transforming the frightening—communicating about death—into the mundane—a conversation with family or friends over dinner—by creating a familiar and comfortable space to begin discussing preferences for end-of-life care and final arrangements before it is too late [21]. One study found that people expressed a desire to talk about dying, but they wanted someone else—e.g., a family member, physician, or friend—to initiate the discussion [22]. DoDs do exactly this—they allow a person to gather family and friends and initiate a conversation people may be uncomfortable initiating themselves.

Anyone with a computer and internet connection can access the DoD website. Thus, virtually anyone can host a DoD conversation. The free and open-access website, in which the researchers have no personal stake or connection, is maintained by a Seattle-based web design company called Civilization and the content was created by Michael Hebb in collaboration with a number of people, including academics, health care providers, artists, and health care CEOs [21]. When one goes to the DoD website, he/she completes a short questionnaire that asks about the person’s goals for hosting a dinner. Goals or reasons for hosting a dinner range from wanting to discuss end-of-life issues because the host, or the host’s loved one, has a terminal illness to believing that having difficult conversations can be liberating [21]. “Homework” assignments such as TedTalks or short news articles are suggested based on these goals, which the host can share with dinner guests to read or watch before gathering for dinner as a way to get guests to think about end-of-life issues [21]. The site also provides language that the host can use in email invitations to guests, which make the purpose of the dinner—to discuss death and dying—clear. Once guests are gathered, the host acts as both a facilitator and participant, and the conversation is allowed to flow according to the topics that interest the facilitator and/or the dinner guests.

This study uses the DoD conversation as a framework for exploring conversations about death and dying because it provides an informal space for discussing what many consider an uncomfortable topic. In this study, the dinner conversations were conducted similar to focus groups. In a DoD conversation hosted by a lay person, the facilitator would participate in the discussion along with dinner guests. In this study, however, the host acted as a conversation facilitator and only joined the conversation to ask a prompt question if the conversation waned.

To date, no studies have empirically analysed DoD conversations. Hence, the aim of the study was to gather friend and family groups to engage in DoD conversations for the purpose of understanding how these groups communicate about death and dying, and whether the DoD approach offers a useful framework for having these conversations. The over-arching research question guiding this study is: How do friend and family groups communicate about death and dying in DoD conversations? To this end, we identified prominent themes that emerged during DoD conversations.

## 3. Grounded Theory

This qualitative, descriptive study uses a constructivist grounded theory approach [23] to understand communication in DoD conversations. Grounded theory uses a systematic approach that guides qualitative researchers to constantly and reflexively code the emerging interactions of interview participant responses. As noted by Glaser and Strauss, this is a process that continues throughout the investigation, from beginning until the end [24]. Additionally, a constructivist approach concedes that our relationships and perceptions are built in and through our interactions with others. When utilizing a constructivist grounded theory approach, the researcher is a part of the research process and a part of the research product. Thus, concepts and themes emerge from the data; however, this approach acknowledges that these concepts are the researcher’s interpretation [25].

## 4. Methods and Materials

The study used the DoD framework to conduct informal, unstructured focus groups among friend and family groups. The following section describes the study’s research setting, procedure, participant recruitment and inclusion criteria, and analysis.

### 4.1. Research Setting

The DoD dinners took place in three primary settings: a participant’s home, a research assistant’s home, or a public restaurant. In public settings, other diners may have been nearby during the conversation; however, in dinners held in private homes, the participants and research assistants were the only individuals present.

The research assistants scheduled the meetings and arranged the meeting place (and/or dinner plans). Dinners hosted in homes were potluck style and participants were invited to bring a dish to share, if they wished. Participants were not required to bring a dish, however. For dinners held at non-meal times, coffee, tea and a snack (e.g., crackers and cheese) were provided by the facilitator. Dinners hosted at restaurants were paid for by the participants themselves, and participants were notified ahead of time that each participant would be responsible for paying for his/her meal.

### 4.2. Procedure

The basic format of the dinners was similar to a focus group; however, the dinner conversations were unstructured. Participants were free to determine which topics they wished to discuss, and the facilitator only used general topic prompts if the conversation waned. Staying true to the DoD design, participants were free to choose when they contributed to the conversation. The DoD design allows dinner guests to share as much or as little as they wish. Facilitators did not call on particularly silent participants and ask them to speak out of respect for their choice not to contribute. Additionally, dinner conversations were transcribed by research assistants who did not facilitate that particular dinner and, during transcription, pseudonyms replaced participant names to ensure confidentiality.

### 4.3. Participants

To qualify for this study, participants had to be 18 years or older and they were recruited by the research assistants. The primary means of recruitment was network (or snowball) sampling. Research assistants asked friends and/or family members if they were willing to participate in a DoD. Given the relationship of participants to the research assistants, every effort was made to avoid coercing participants to participate. Invitations to participate in a DoD conversation used the stock language provided by the DoD website; however, they also included information that the event was for research and participation was voluntary. The consent form was attached to the email, and prior to the conversation, consent forms were reviewed in depth, reiterating the voluntary nature of participation. Data were collected until saturation was achieved. As noted by Strauss & Corbin, saturation occurs when the coding yields no new information [26].

### 4.4. Analysis

Emergence is at the heart of grounded theory, and this study used inductive thematic analysis [24,25]. More specifically, the data were analysed manually using the six-step thematic analysis technique outlined by Braun and Clarke [27]. First, both authors familiarized themselves with the data by reading and re-reading the transcripts and writing down initial ideas. Next, both authors generated initial codes independently. After the initial codes were discussed, the authors searched for themes independently and reviewed the themes together. Working together, the authors completed the fifth stage of the analysis, defining and naming themes, entailed “identifying the ‘essence’ of what each theme is about, and determining what aspect of the data each theme captures” [27] (p. 92). The last step of the outlined thematic analysis technique includes writing the report. In an effort to reduce bias, only the research assistants facilitated DoD conversations. Thus, the co-authors did not participate in any of the dinners they coded and analysed.

## 5. Results

Given the sensitivity of the topic, the researchers believed that the group setting of DoD conversations allowed participants to participate as much or as little as they wished, and the informality of the dinner conversations shifted control from the facilitator/host to the participants/guests, allowing all who participated to ask each other questions and probe for more information while also sharing their thoughts, ideas and experiences.

Following institutional review board (IRB) approval, 52 DoD conversations were conducted. The dinners were facilitated by the researchers or research assistants who were extensively trained and IRB-certified. Facilitators began by reading a short welcome note and reviewing the consent form, which each participant signed prior to participating in the DoD discussion. All participants were informed that they could leave the conversation at any point if they wished, and their contributions would be redacted. Of the 240 participants in this study, none chose to withdraw from the study. Conversations were allowed to develop organically; however, facilitators had unstructured question prompts they could use if the conversation waned. The length of the dinners ranged from 32–184 min, and all of the dinners were audio-recorded. Research assistants transcribed the recordings. Once the dinners were transcribed, the audio recordings were erased.

Each dinner consisted of 4–8 participants and resulted in 240 participants ranging in age from 18–76. There were 109 men (Mean age = 27.85) and 118 women (Mean age = 31.00) who participated in the study. Thirteen participants indicated that they were transgender or selected “other” on the demographic questionnaire. Of the 46 DoDs, 27 were friend and/or colleague groups and 19 were primarily family groups. Participants were recruited from a metropolitan area in the mid-western states of the United States of America (USA).

The analysis revealed that there were more similarities than differences in how friend and family groups communicate death. One difference that did emerge was that in friend groups, which also included work colleagues, the participants were more likely to state that they were unable or uncomfortable talking about death with family or that they could be more candid talking about death to friends than family members. Participants in friend groups would often report “I would never tell my family this”, “my family would never approve of this”, or “I think it’s difficult to talk with family about it”. The opposite statement was not made of family groups about their friends.

The analysis also revealed three main themes (with various subthemes) that illustrated how participants communicated about death: Desire for a Good Death, Tactics for Coping, and Topics that Confuse or Elicit Fear. (Note: in the following paragraphs, direct participant quotes are cited with the DoD identification number first and the transcript line numbers second. For example, 123: 45–46 would be DoD number 123 and transcript lines 45–46).

### 5.1. Desire for a Good Death

The first theme relates to reported desirable and undesirable outcomes when it comes to what people want at the end of their lives. When asked what they perceived to be a “good” or “acceptable” death, depending on the facilitators’ word choice, participants were more likely to first report what they did not want at the end of life before they were willing or able to elucidate how they would prefer to die.

#### 5.1.1. Dreaded Death

When describing their perception of a “good” death, participants were most likely to first respond with what they did not want at the end of their life. Many participants responded that they “don’t want to be a vegetable” or did not want to “live on machines with no hope of recovery”. Also, many participants reported that they “dread drowning” or dying by “fire.” As one male in his 40s succinctly stated, “I don’t wanna drown. And I don’t wanna burn” (230: 481), and his brother retorted, “Yeah, true dat” (230: 482).

The most notable theme related to what participants did not want at the end of their lives was to “be a burden”. However, when probed by DoD facilitators, most participants were unable to explain what would differentiate as burdensome versus unburdensome to their families. Many expressed that the mere dependence on friends or family members at the end of life was more than they could bear. However, many family members and also friends retorted that it would not be a burden and would, in fact, be an honour to take care of them in their time of need. Also, the mention of not being a burden was especially prevalent among men:
Man 2:But I personally, speaking for myself, would feel like I am a burden.
Man 1:I think that is a pride thing for anybody.
Man 2:I don’t want to have to have my butt wiped.
Woman 4:I think it is harder for a man to be dependent than for a woman.
Woman 5:I agree.
Man 3:I can see that.
Woman 1:Because men are more about not burdening their family and women are, like, well, it would be nice to be able to take care of you and see you in your final moments, in that kind of way.
Man 2:But, I feel like if I am at the point where I can no longer provide and protect my family, you know, as a man, I would want to go home with God (230: 497–507).

Additionally, many participants identified an untimely death as a dreaded or even feared death. Untimely death was discussed as one that comes before a person has the opportunity to accomplish what he/she/they wishes to accomplish. For example, a male participant said:
Man 1:I fear what things I haven’t done that I should have done.
Woman 1:Haven’t finished.
Man 1:Yeah, finished up. And If I don’t feel I like I got anything done or finished am I going to come back as a spirt or you know.
Man 2:Mmm-hmm.
Man 1:That’s basically it. That’s how I feel about that (123: 94–100).

#### 5.1.2. Desirable Death

When expressing what they desired at the end of life, participants’ descriptions were somewhat more vague and ambiguous. For example, participants said that they would prefer their death “to be inexpensive for the family”, have a “quick and painless death”, or to “go out with a bang”. Many participants expressed a desire to “die peacefully in their sleep”. As one father, humorously expressed in a family DoD:
Man 2:I would prefer to be in my sleep.
Woman 1:Yes, I would agree with that I would want to be in my sleep too. I would like to die comfortably, quickly, not painfully or long term (232: 60–63).

If being a burden to one’s family was considered an undesirable death, having some quality of life before or during death was described by many participants as a desirable death. This related to people’s wishes to have painless and quick deaths. However, how participants described quality of life in the conversations varied. Some participants referred to it generally, simply stating that they wanted to have “quality of life” at the end of their lives or “quality over quantity” of life. Other participants described it more specifically as being able to do some of the things one enjoys. For example, one woman said, “I feel like as long as I can read and not be bored out of my mind, I would be okay dying” (134: 159–160).

Another participant conceptualized a desirable death as one that was positive for survivors: “Well, it seems to me that a good death is when the people that survive you, uh, have a role to play in your care, and uh, see you at your best at the end, whatever that happens to be” (119: 280–281). This also reflects participants’ insistence that caring for the dying is an honour and privilege, despite fears that it makes one a burden to family and friends. Related to this was a desire to be remembered, which was expressed by some participants. In the words of one man, “I want to be remembered, I mean everybody wants to be remembered” (124: 209).

### 5.2. Tactics for Coping

As noted earlier in this article, discussing death can be uncomfortable. The analysis revealed that participants in this study used several coping tactics for dealing with the discomfort of discussing or thinking about death. Coping tactics included communication strategies that moved the conversation away from tension or uncomfortable interactions; invoking one’s spiritual beliefs to reassure or comfort them when talking about death, or separating oneself from death or aging in order to not think about it.

#### 5.2.1. Humour to Diffuse Tension or Deflect Discomfort

Regardless of friend or family group, humour was prevalent throughout the DoD conversations. The humour was used in two ways. First, it was used as a release valve for a difficult part in the conversation to diffuse tension. For example, in one family DoD when a mother and daughter engaged in an emotional exchange in which the daughter told her mother to complete an advance directive because she (the daughter) would be “selfish” and choose to keep her mother on life support indefinitely, another family member interjected and said, “How about them cowboys?” (130: 607), after which everyone laughed and the tense moment was diffused.

Second, humour was used to deflect the discomfort of thinking about death, particularly preferences for final arrangements. For example, one male mentioned that he wants his funeral “to be a party, and I want to be taken out in the cheapest pine box available” (131: 94), and a female queried whether “Viking funerals are still allowed” (121: 168). Another male dyad quipped to each other “you can just leave me out in the garbage” (135: 196), and “I just want to be composted” (135: 196). Participants were particularly creative when describing what they wanted to have happen to their bodies with one male saying he wanted “my femur turned into a sword” (131: 114), another male mentioned that his family can “skin me and turn me into a football” (121: 373), and a female asking if she could “do like an Eskimo version, just push me out on a block of ice and let the polar bears take care of it” (120: 525–526).

#### 5.2.2. Spiritual Reassurance

A vast majority of participants took solace in their spiritual beliefs when it comes to death. Many felt that it was important to take care of the issues related to their earthly life, but felt that everything would work itself out and they would be happy and saved by a higher power. As an older female mentioned, “if I die, according to scripture I feel great that I would be in paradise” (125: 280–281). Another female participant explained that her amalgam of religions give her solace:
Yea, so like I said earlier I’m Romani but I’m also part Native American, which you would never tell by my skin tone but oh well. And I’m also Wiccan so, uh, and being raised in a Roman Catholic setting just, the spirituality has always been in me so I know what the Roman Catholics believe, and what the Wiccan believe, and what the Romani believe, and what the natives believe, and I just kinda mix it all together and it gives me a real strong sense that there is, uh, a better place out there. Like it’s not gonna be worse than this, and it’s not just gonna be this. There actually is a place better than this.(126: 406–416)

In contrast, individuals who did not express a spiritual association were sometimes envious of those who did. As one young female expressed:
For me like, death is inevitable, there’s nothing we can do to get away from it. I know that’s ugly and scary. It’s natural to be scared of things like that we don’t understand or, like, the “unknown”. But at the same time, like for me I don’t have faith either, so, like, we gotta make the most of it while we’re here ‘cause this is all we have. For me, I envy people with faith ‘cause it would be nice to, like, believe in something on the other side, something for you, something better.(126: 399–403)

#### 5.2.3. Separating Oneself from Thinking about Death

Many participants expressed that “I don’t really think about [death]” (221: 152), or “death doesn’t really bother me, um, I never really put a lot of thought into it” (224: 90). These participants tended to be younger. Most admitted that they had thought about their parents’ (or especially) their grandparents’ death, but the likelihood of their own death was beyond their grasp. As one young female mentioned:
Because I don’t see myself growing old. Like I really don’t see myself growing old. So like dying of an illness or dying of natural causes doesn’t seem like something that’s going to happen to me.(223: 161–163)

This theme often overlapped with the previous theme as many participants who confessed that not knowing is the best way of knowing also cited a spiritual preference, or lack thereof, as the impetus of their not knowing. One young female participant illustrates this in the following quote:
Well I would argue, as the person in the room without any faith (if you weren’t aware), as the person in the room without any faith, I don’t worry about it. I was just fine before I was born and I’ll be just fine after I’m dead. I won’t know the difference.(228: 104–106)

### 5.3. Topics that Elicit Fear and/or Uncertainty

The third theme describes the topics that elicited fear or uncertainty among participants. The main sources of fear and/or uncertainty issues related to organ and whole-body donation, hospice and palliative care, and wills and advanced directives.

#### 5.3.1. Organ and Whole-Body Donation

Although a majority of participants expressed that they thought organ donation was important, many rebutted those notions with negative comments. As a middle-aged female revealed, “it’s like if you were in a car accident and pronounced dead for two minutes, I don’t want them to just rip them [internal organs] out or anything” (131: 209–210). There were many examples of participants bantering back and forth about organ donation. First, two young females dialogued with a male who thought that medical professionals were likely to end an organ donor’s life early because of their donation decision:
Woman 1:I don’t like it to be known [organ donation], because what if somebody was like….
Man 1:Think they kill you off quick just to get your organs.
Woman 2:That is a genuine concern for a lot of people (130: 780–784).

Another woman in her late 50s expressed her fear that organ donors, if they ended up in the hospital, are misled by health care providers:
They might think that they are saving their life and bringing ‘em back later, but they can harvest their organs or whatever, which is so someone can use it.(123: 233–235)

In addition to some participants feeling wary of organ donation, some participants also felt that donating their bodies to medical science was financially detrimental:
Man 1:But I found out through my insurance agent that one, it costs a lot of money to donate your body to science.
Woman 1:What?!
Man 2:To donate it? (134: 225–228).

#### 5.3.2. Hospice and Palliative Care

Although many participants expressed positive feelings about hospice, hospice and palliative care were also topics that elicited uncertainty among the participants in this study. Those who had experience with hospice often spoke authoritatively about what hospice does and does not do. As a woman in her fifties expressed:
That’s why we brought my mom home. We didn’t put her in the hospice for that reason. Until the day they said well… her lung was filled up with fluids and they [hospital staff] said… “Was she in hospice?” And I said, “No”. And they said, “If she was in hospice they would leave that fluid in there and you would sit there and watch her suffocate”. So we, I said, “No, we don’t have that she’s not in hospice”. “Well you can sign it now”. I said, “No I don’t, I’m not signing it”.(123: 214–218)

Another man held that hospice does not provide antibiotics and that by choosing hospice, he was choosing death for his parents:
My mother actually called the hospital at one point to have dad taken to the emergency room and hospice found out and wanted him sent back home, because they are not trying to do anything that is helping keep him alive basically. Until I had the personal experience I never realized what all was included in going to hospice. Even if you get a cold, you do not get any antibiotics, they let it go into pneumonia.(135: 488–495)

#### 5.3.3. Wills and Advanced Directives

The use of a will is complex. Many of the younger participants talked about the fact that they didn’t care “who gets my blender” (221: 56) or as one young women illuminates, she doesn’t feel the need to complete a will or advanced directive because “I don’t even have anything to give, like who gets to keep my cat” (128: 206–207). In the following excerpt, a man in his late 20s explained when he thinks someone should complete a will:
That’s one thing like I don’t like about wills. I mean, I don’t want to have a will. Like if I had a will, it’s going toward something, like, positive… If I have children I want that to go towards their college… I’ll make sure that goes towards their college or anything…and like not want them to have it to waste or anything like that. But, like, if I’m single…like if I’m like a forty-year-old man or something like that dying early, like an early death or something like that, and I have a will… I’ll give it to charity or something like that because… I believe a will should be…your final thoughts with your family. Like you wanted them to know this, like, before you died, and it’s like it shouldn’t be a material thing. It should be a closure.(127: 446–455)

One middle-aged female talked about how making end-of-life decisions for someone else is still difficult, even when there is an advance directive:
It’s still hard because you got a piece of paper and possibly have to let go of someone you love or make a decision not to give them food, and you still have to watch the result. So, um, that’s why I think, yeah, yeah, you’re trying to honour their wishes, but at the same time...(129: 217–225)

Participants also had some uncertainly about their family members’ willingness to honour their wishes, even if they were to express them through advance directives. As two participants noted, “I don’t think they would, or take me off it [life support] I think they’d put me on for as long as I can go,” and in response, “Yea I feel like our parents would probably fight for as long as possible” (125: 208–209).

## 6. Discussion

This study sought to understand how family and friend groups communicate about death and dying during DoD dinners and whether the DoD approach offers a useful framework for having these conversations. The analysis revealed that friend groups displayed more candidness than family groups when discussing death; overall, how family and friend groups communicate about death and dying in the DoD context is more similar than different. While it was the researchers’ initial thought that there would be substantial differences between the communication in family and friend groups, those differences were not evident in most cases. Many family groups did not talk about specific family instances but rather their own views of death. Spouses and/or significant other pairs tended to quip back and forth between each other about wishes, but those quips were similar in same-sex and opposite-sex (close) friendship pairs.

The only difference that the analysis revealed was that participants in friend groups said that they were being more candid than they would be with family members. The candidness among friend groups may be the result of the study sample. Participants in this study were younger adults, and younger adults tend to disclose more to friends than to family members [28,29]. This may be problematic when it comes to end-of-life decisions as friends have no legal rights, but may in fact know what the person actually wants at the end of their life.

From the analysis of over 240 participants, it is clear that there are many misunderstandings about death that have yet to be dispelled and are quite prevalent in contemporary discussions about death. The results of this study highlighted topics that made participants uncomfortable or fearful, which suggests that the participants have some cognitive contradictions about death. The results revealed four contradictions. First, participants whole-heartedly trusted that family would equitably distribute personal items (especially if a will did not lay out the distribution of property), but they did not believe that family members could be trusted to follow end-of-life wishes. Although the research on the effectiveness of advance directives varies, studies that suggest the presence of advance directives results in end-of-life care that more closely aligns with a patient’s preferences [30,31].

Second, the topic of wills and advance directives contradicts what many participants said about what constitutes a good death. Participants intellectually believe in wills and advanced directives but felt these documents/processes were not necessarily for them. Participants also wanted their family to be comfortable and not to be burdened by the loose-ties of their lives, but it often wasn’t significant enough for them to consider the need to plan ahead. This is consistent with other research that noted that although people think advance directives, like living wills, are good, they do not complete them [32]. In particular, younger participants expressed indifference with wills and advance directives, which is consistent with research [33,34,35].

The third contradiction was that participants claimed that they did not want them or their loved ones to die in the hospital, yet participants expressed deep distrust in hospice and/or palliative care. Based on their expressed concerns with hospice, participants were clearly uninformed about the differences between palliative and hospice care. Whereas palliative care, which focuses on quality of life and the whole patient, is available to people at any stage in an illness, hospice care is only for people with a terminal illness diagnosis, a life expectancy of six months or less, and who have accepted palliative (for comfort and pain management) instead of curative care [36]. This misunderstanding is not unusual, however, and Cagle et al. found that although many of the participants in their study had heard of hospice, many of them were unaware of the parameters of receiving hospice care [37].

Lastly, contradictions related to organ donation were prevalent. Participants thought that, intellectually, organ donation can be seen as a societal good, but they did not trust the organ donation process. This mistrust may be the result of mass media representations of organ donations [38], which tend to portray the organ donation process as negative and morally corrupt [39].

Although the results reveal several contradictions related to participants’ desires, perceptions, fears, and uncertainty about death and dying, the DoD format provided a space for participants to share their thoughts, feelings and experiences. Overall, participants communicated various feelings about their DoD experience including: “thought-provoking”, “overwhelmed”, “confused”, “mind-blowing”, “a little depressed”, “the same”, “empowered”, and “relieved”. Moreover, a majority of participants found the experience positive and noted similar sentiments to a participant who claimed that “I think we just have more of this. I think having more conversations about it makes it less hard to take” (133: 942–943). Thus, although the communication that occurs in DoD conversations is at times contradictory, for many it is a positive experience nonetheless.

## 7. Limitations

This study was limited in three ways. First, we relied on a convenience/snowball sample. Although this sampling method in and of itself is not overtly negative, in this case it could be that family and friend groups that agreed to participate were more open and willing to talk about death. Second, participants were aware that the dinner conversations were part of a larger research study, which could have influenced what they said. Third, a majority of family and friend DoDs were conducted in one geographical area. As a qualitative project, our goal is not to generalize to other audiences, it is important to remind the reader that the project is descriptive and not prescriptive. In the future, it will be important to expand the study and examine end-of-life discussions nationally and internationally.

## 8. Conclusions

This study asked how DoD participants communicated about their end-of-life choices and also empirically analysed DoD experiences. The results revealed three prominent themes that at times contradicted each other; however, given the importance of communicating about the end of life, the conversations provided a space for participants to share their experiences, feelings, fears and hopes.

Hosting a DoD among family members allows participants to share their preferences with those who may become their surrogate decision makers, whereas DoDs involving friends may provide a context for exploring one’s thoughts about death or expressing concerns not easily shared with family members. However, DoDs may also serve to perpetuate misinformation as friends and family members can express uncontested untruths during the dinners and, as a result, may contribute to people’s anxiety about death. Despite the potential drawbacks, however, based on predominantly positive feedback participants share, DoD events offer a promising method to encourage people to talk about their end-of-life wishes and feelings about death.

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
