# Peer review of "Contradictions and Promise for End-of-Life Communication among Family and Friends: Death over Dinner Conversations"

_behavsci, 2017, doi:10.3390/bs7020024_

Round 1
Reviewer 1 Report
The paper is about conversations and death and dying – it uses thematic analysis as described by Clarke and Braun to identify themes in these researcher-led conversations.
Please find some concerns below, further detailed further down with reference to specific sections.
There is not a research question/goal in your abstract.
The paper also lacks a clear research question. “Learning about how DoD conversations function” is very general. “Function” is not defined. Could you please break it down into clearer objectives? Also, if your focus is on how these conversations “function” then you would look at how they are structured as activities (e.g. through conversation analysis) whereas in the findings you mostly report on the contents of these conversations (the only example being perhaps the use of humour).
Goal/methodology mismatch: as currently described the paper is about how people talk about death and dying with the support of an online tool. However, we later find out that researchers or research assistants led these conversations. So, people where asked to discuss death and dying by researchers. This makes it likely that they treated these conversations as interviews. At the end of the results, you comment on how people talk about death and dying differently depending on whether they are with family or friends. However, you do not take into account the fact that they were discussing these topics in the presence of researchers and following prompts/questions designed by researchers. I wonder if it would make more sense to simply say that you interviewed people regarding their views on some issues regarding death and dying. Also, you do not describe the DoD tool – is it conceived for people to use on their own or does it always require facilitators. And did you modify this approach by embedding the intervention into a research project and having researchers facilitate the conversations? If so, how? To summarise this point: there currently is a lack of fit between your stated goal/focus and chose methodology. Could you please amend or alternatively justify more your selection of the terms you use? And importantly, could you clarify if your participants were visibly treating the conversations as interviews? I think this is crucial because it limits the transferability of your claims; i.e. you cannot extend your findings to how people discuss death and dying outside the contrived context of a research interview.
It is hard to find coherence in the study findings: the themes are somewhat disjointed and it is hard to find a core finding. It would be good to articulate this more strongly. Related to this, please explain more clearly what we already know and what the study adds.
Could you refer to COREQ or other internationally recognized checklist for qualitative studies? There is missing information on your methodology (see below).
Why did you decide to use the DoD approach? Do you have a stake on it? Did you design it? Why use this approach and not another one?
“DoD does this by transforming the 59 frightening into the mundane by creating a familiar and comfortable space to begin discussing 60 preferences for end-of-life care and final arrangements before it is too late” This sounds partisan. I would encourage you to adopt a more impartial stance and consider the outcomes of the tool as an empirical problem rather than praising from the outset. E.g. whether it makes people comfortable or not is their own judgment and a matter of empirical inquiry.
More detailed comments:
2.1 does not describe how the internet-based tool was used. Your stated goal is about how DoD conversations function. But here this is completely lost. I would expect a clear description of how the DoD tool works and how you used it to conduct the conversations. Tables and appendixes could be used for this.
Refer to COREQ or other checklist
p.3 line 94 – results not requiring deep analysis – sounds problematic; one would expect all findings are based on deep analysis
p. 3 lines 95-97 – sentence unclear
2.1 Please provide the interview/topic guide in a Table or Appendix. It is unclear, when we get to the results later, how those topics emerged (e.g. organ donation). Those topics are quite specific and not necessarily things that people would discussed if not prompted (e.g. hospice, advance directives). See COREQ checklist
Use of quotes from interviews: can you track participants with anonymous labels so that it is clear if the same person is talking in more than one quote? Is this the number you use after your quotes? See COREQ
When you use quotes (A man said X; A woman said Y) can you clarify the extent to which these are representatives of trends in your data?
For each subtheme can you make explicit the context of participants’ responses? What did you ask them?
3.1 If this is a category, it requires definition before moving on to sub-themes. Please define the category and how the particular sub-themes relate to each other and sit well together within the category. Consider adding a diagrammatic representation in a figure. Most importantly, is this a tactic for coping with what? This should be in the section heading. Later you report that participants do not give death and dying much thought (you explain this by noting they are young). So, does it make sense to say they are coping with death and dying if it is not a relevant concern for them? Or does ‘coping’ refer to something else? Could you clarify?
3.1.1 Humour as deflection of what? Also, here you merely report that participants made humorous remarks. Can you bring more depth to this observation? For instance, in what circumstances did they use these humorous comments? Additionally, I would expect you to make sense of this finding later in the discussion section.
3.1.2 Does ‘religious’ capture your participants’ perspectives? Would ‘spiritual’ be a better fit? The heading suggests that people are reassured by religion but it turns out that for some this is not the case.
line 129 The sentence seems incomplete.
3.1.3. ‘Ignorance is bliss’ seems more your judgment than a participant meaning. If the interviewees said that they do not think about death, they did not convey that this a source of bliss nor did they characterize their state as ignorance. Could you take a more emic/endogenous perspective? Or alternatively justify in the methods section why and how you can adopt an external perspective in labelling participants’ expressed meanings?
3.2 I find the same problem as in section 3.1.3. Who decides what is a misconception? Did the participants themselves label their own perspectives as ‘misnomers’ or did you? This points to a methodological problem in the paper which you need to solve. You either stick to what your participants conveyed and only evaluate them in the discuss section; or alternatively you take an explicit evaluative stance. In the latter case you need to clearly justify the basis for evaluating your participants’ views – for instance by making explicit what your view/philosophy is regarding death and dying matters. These are matters that attract conflicts of values, so if you set out to evaluate whether your participants are right or wrong about any given issues, you need to provide a clear basis for doing so. This will, in turn, make your procedure transparent.
Heading title ‘Topics that confuse’ – can you define confusion? Confusion in what? Again, did you decide your participants were confused or did they say they were confused? Also are the categories you use representative of all your participants, some, most of them?
3.2.1 The heading is not informative – it should not just be about the topic being discussed (e.g. organ donation). I suppose this was already in your topic guide for the interviews, so the fact that it got discussed is by definition not a finding. The finding has to do with the meanings expressed in the interviews. Here people seem to express concerns/fears. What is their nature?
3.2.2 You discuss confusion. Would this be better characterized as lack of knowledge?
4. What do you mean by ‘paradox’ and how does this term apply to your data? As compared to other possible terms such as tensions, conflicts, contradictions, dilemmas, ambivalence, and so on. I invite you to define your terms and concepts, and to support their use. You point to a mismatch between general opinions (organ donation is good) and more personally relevant perspectives (I would not donate my organs). Is this surprising? Why is it a paradox? If a paradox is a statement that leads to a self-contradictory or a logically unacceptable conclusion, where is the contradiction here? And do participants experience it as a contradiction?
Line 280 – why and how would DoD perpetuate misinformation?
Many thanks for giving me the opportunity to review this paper. I hope my comments will be useful to improve it and that the study will be published.
Author Response
We have responded to both reviewers in the same comment matrix. The web portal only allowed us to submit one document at a time. Our matrix and the interview protocol are attached. Please let us know if you need additional information or attachments outside of this portal.
Attachments include:
Edited manuscript
List/matrix of all changes requested (Reviewers 1 and 2) and our comments to those requests also the semi-structured protocol that guided our DoD conversations is included. We have also sent our COREQ checklist to the primary editors.
Best,
Andrea and Jessica

Reviewer 2 Report
Thank you for the opportunity to review this manuscript. This is an important issue and the DoD movement is having quite an impact worldwide.
This manuscript is easy to read and flows nicely. However, at times it reads more like a newspaper article or editorial than a scholarly piece of writing. There are numerous areas where it can be improved.
Introduction
The issues with increasing rates of death and increased hospitalisation for dying is not an issue only for the USA. Rather, the situation you describe is common across many developed countries where there are rising rates of chronic illness and rapidly ageing populations. That said, I think your introduction is too USA-centric. Given that academic journals these days are read by worldwide audiences, I urge you to re-frame your introduction to me more cognisant of this. Perhaps you could provide some international context by referring to the World Health Organization work on ageing worldwide and then cite various international statistics on death and where it occurs. For example. In Australia, the statistics on in-hospital death and preference for place of death and almost identical to what you report in your introduction, so why not include international evidence.
In the second paragraph of the introduction, I suggest you remove the ‘they’ from the first sentence as it is superfluous if you already have ‘he/she’
At the top of page 2, you include a very long quote starting with ‘timelier referrals…” I suggest you paraphrase this as there is no need for it to be quoted verbatim. Paraphrasing better demonstrates your understanding and is more scholarly. It will also mean you can change the upper case Ds on Dying to lower case. Save the quotes for seminal work.
Next paragraph, I do not understand what you mean by ‘people need to develop communication competencies for managing the end of life’. Which people? Typically the term ‘competencies’ relates to an assessment, such as when you deem a person competent in a skill. So it doesn’t seem right here. I am not sure if you are trying to say ‘In order to facilitate communication about death, people need skills and confidence in communication’?
Same paragraph, I do not see why you need to quote “Normalize conversations about death and dying”. It is such a short phrase, and given that there is little other ways to same the same thing, I would write it as is, but without the quotation marks and page reference. They are unwarranted. Just cite it as normal.
With “about care goals, and preferences related to advance serious illness”, please paraphrase
The sentence commencing with ‘DoD does not promote a particular EOL care preference….’ This sentence must be referenced.
Next paragraph, please remove ‘Overall’, rather just commence with ‘In this study’
Replace ‘death over dinner’ with DoD. Once you introduce the acronym, you should use it consistently thereafter.
What do you mean by the following sentence?:- ‘Since DoD conversations have not been empirically analysed the impetus of this study was to uncover general themes that emerged from….’ This sentence is not clear. I think what you are trying to say is...’To date, no studies have empirically analysed the DoD conversations, hence the aim of this study was to….’
What was the AIM or PURPOSE of this study?
2 Materials and Methods
Can you please include an opening statement/s as to the methodology used here and why? Eg qualitative descriptive study
Was it a qualitative descriptive study? I cannot be sure because I cannot find much to tell me about the survey. Was it a descriptive survey, did it contain open-text questions or scale or binary (yes/no) questions? Where did this satisfaction survey come from? Was it previously validated? Is this part if the study reported here? If not, then suggest you remove reference to the satisfaction survey in section 2.1
2.1 Procedure
What do you mean that conversations were ‘collected’?
Where these DoD going ahead anyway and your team were given permission to record them? Or did your team set up the dinners? If they were going ahead anyway, how did your team of researchers get involved? How did you find out about the dinners? How did you get involved?
2.2 Participants
Please replace ‘Mage’ with ‘Mean Age’ for clarity
240 participants – did anyone at the dinners choose not to participate? If so, how was this managed? Or did you require a whole of dinner consent?
Again, how were the DoD groups formed and what role did you research team play in the formation of these groups?
How did you record the dinner conversations? I assume you audio-recorded them but how did you manage to stop people talking over each other? Did this impact on the quality of the data?
2.3 Analysis
Who transcribed the audio-recordings to transcripts?
When, during your analysis, did the two authors compare their analysis they did independently and how were discrepancies resolved?
3 Results
Please use third person language.
The first sentence is talking about the process of analysis. This does not belong here.
All of the small quotes used here should be referenced according to who said them, or at the least, which interview they were in. For example:- “don’t want to be a vegetable” (DoD 5) representing that this quote came from Death over Dinner number 5.
I actually think this first paragraph should be a theme, not an introduction to the rest of the themes.
3.1.1.Humor as Deflection
Thank you for including references to where the quotes came from but you need to include a sentence at the commencement of the results section to explain to the reader what it means. For example what does 130: 607 mean?
3.2.4 Family vs. Friend Group Differences
Whilst this is very interesting, you have mixed the reporting of results and ‘interpreting’ or ‘discussing’ them here. For example, ‘Friend groups tended to be more candid that family groups’ and ‘As you can imagine the opposite statement wasn’t made of family groups about their friends’ – these are not results. Similarly you should not be referencing other materials here in the results. You should just report results here, and then move the interpretation, discussion or commentary to the ‘Discussion’ section.
4 Discussion
Please review for change in tense.
A sub-heading of Limitations would be helpful. I don’t see how or why snowballing is bad. You need to explain why. How would you do it differently? The issue of being in one geographical area is not an issue in itself, it only becomes one if you try to infer national or international findings from this sample.
Your suggestion that further work is needed for military personnel seems out of place here. The quote from the male in his 20s about being in the military does not belong here. It needs to be presented in the results. You can then link the possibility of further work in this area, but not in a paragraph about limitations.
The quotes at the end of the discussion are out of place here. Put them in the results.
In summary
Overall, I would say that this is very important work. I have a very good understanding of the DoD movement and the power of this work, BUT this paper needs significant refinement before it is suitable for publication. I do trust these comments are useful.
Author Response

(The authors gave the same response as above.)

Round 2
Reviewer 1 Report
I have found the manuscript significantly improved and I recommend publication following some more minor amendments. I would encourage you to address the comments I list below. I find the study very significant and meaningful; I believe it will make a great addition to the extant literature.
Abstract
Please clarify that you carried out an intervention. It must be clear that you hosted the conversations. The link between intervention and descriptive study must be absolutely clear and transparent.
Mention your data collection technique.
Avoid the passive voice – a website “was created” makes it ambiguous whether you created the website.
Problem with the sentence “Our analysis revealed that little difference in how family and friend groups communicated about death and dying in DoD conversation” – please clarify
TYPOS
p. 1
“This make communicating” – makes
“Australian Medial Association” – medical?
“engaging in in it”
p. 2
“making it easier for people [to] prepare”
“people my” – may
p. 5
“ot diffuse tension” – or
p. 6
“fear and/or unDertainty”
p. 7
“many participants expressed positive feeling[s] about hospice”
COMMENTS
Introduction
Could you clarify who the host is in DoD conversations?
“Given the sensitivity of the topic, the researches believed that the group setting of DoD conversations allowed participants to participate as much or as little as they wished, and the informality of the dinner conversations shifted control from the facilitator/host to the participants/guests, allowing all who participated to ask each other questions and probe for more information while also sharing their thoughts, ideas, and experiences.” – This sounds more like results and should not be here. Is this the part where you explain your interest in DoD conversations and why you decided to research them? Can your interest be explained in a way that does not portray you as expecting specific results? (Which is at odds with the qualitative nature of the study). Alternatively, if you were expecting such results could you justify the bases for doing so as well as explain how you made sure that your analyses were not biased by such expectations? Reading a bit further it becomes clear later in the Methods section that you conducted the conversations, so the study was an intervention; in this light it made sense for you to assume that the intervention would be beneficial to participants. Could you clarify this early on in the Introduction? Please also add a sentence to the Methods section on how you made sure your analysis was not biased, given your multiple involvements.
Please clarify who created/maintains the DoD website and that it was not you. It is not obvious from the manuscript.
“We will also discuss the potential benefits and drawbacks of using the DoD conversations as a medium for sharing one’s end-of-life preferences.” – promise not maintained. This is only hinted at in the Conclusion but requires elaboration.
Methods
I would definitely publish the topic guide you attached for reference. Could this be also elaborated upon? For instance, in the results section (3.1.1) you report asking people what is a good death for them, but this is not in the topic guide.
Results
It is not always clear if some quotes are representative or not. For instance, in 3.1.2 “Another participant conceptualized a desirable death as one that was positive for survivors” – were they the only one out of the 240? If so, why did you select this particular quote?
Discussion
“The analysis revealed that friend groups displayed more candidness” – this is currently stated and not shown in the results. It was not clear in the Results whether people SAID that they are more candid with friend; or whether you observed them being more candid with friends in the DoD conversations you conducted. Throughout the paper there are contradictions about this point. Could you clarify this?
“it was the researchers’ initial thought that there would be substantial differences” – I think it is very good that you state your expectations (see my earlier comments on this) but please report these earlier in the manuscript.
Sentence about ‘misnomer’ at lines 361–363. This seems disjointed from the surrounded text and out of context. It needs elaboration.
Cognitive contradiction – please define this expression. Subsequently you seem to explain some of the contradictions in terms of people’s lack of information or misconceptions. If this is the case, could you clarify this above where you introduce ‘misnomers’? Would the term ‘misconception’ be more suitable? You should make it clear, perhaps in the Methods, that here you take an external perspective and do not merely report your participants’ meanings – you comment upon them as ‘misconceptions’. This is an additional step in your analysis process. This is fine but should be methodologically justified. I had raised this in my previous review. Currently, section 2.5 is very generic – perhaps you clarify the practical steps you undertook.
Your examination of cognitive contradictions: these are findings. They look like a higher level of analysis connecting and comparing meanings across thematic areas – this step should also be mentioned in the Methods and reported in the Results. The Discussion section should deal with other things (for example, academic significance, implications for society, directions for future research).
People’s positive feedback on DoD: these are also findings, not elements of Discussion.
Currently, there is no real Discussion in the manuscript. What do we learn? What are the implications?
“DoDs may also serve to perpetuate misinformation” – how so? In your rebuttal letter you say you have removed this but it is in the Conclusion.
Author Response
Thank you so much for your willingness to review the manuscript again. We hope we have sufficiently addressed your concerns. The matrix we have attached includes the edits/responses for both you and Reviewer 2.
Best,
Andrea and Jessica

Reviewer 2 Report
Thank you for the opportunity to review this manuscript again. The manuscript is improved since the last submission however there are still a number of issues that need to be addressed.
You use end of life and end-of-life throughout the manuscript. Please choose one and use it consistently. My experience is that the trend is towards using the term without hyphenation.
ABSTRACT
line 15 spelling of resources
line 18-19 please match this question with what is written in the body of the manuscript
INTRODUCTION
Line 41 in in
Line 57 about
Line 66 may
Line 78 researchers
MATERIALS AND METHODS
Line 92 should say grounded theory
PROCEDURE
Line 111-114 - this information, commencing with 'Of the 240 participants..' belongs in the beginning of the results.
SAMPLING
I am left with concern on how the recruitment actually happened. It reads as if the authors rallied friends and family, and their friends and family for this study. Were issues of coercion considered? For example, if you invited family to participate, how could they say not? Were issues of 'existing relationships' considered? how did the researchers separate family relationships and friendships from researcher-participant relationships? I am also interested in who paid for the dinners? If the research team funded the dinners, could this incentive have further coerced people? you have not mentioned who paid for the dinners at all, and I think it shoudl be overtly explained to allay any assumptions.
Line 124 you state that 'Data were collected well after saturation was achieved..' Whilst your definition of data saturation is correct, reference to it does not belong here. This should be reported in the results or analysis section. Furthermore if you are going to say that data collection continued after data saturation was achieved, then say why. Typically I would expect that the point of data saturation becomes the point at which data collection ceases. So why did you keep going?
Line 127 this paragraph is again reporting results. Typically the beginning of the Results section is where you would report demographics of the population, length of interviews etc.
How do you define family groups? People related by blood? people who co-habitate? or did you give participants the option to identify as family or friend groups?
RESULTS
Please add the info from the Procedure and Sampling sections above to the results section
Line 167 the should be their
Line 196 'would prefer for their deaths'
Line 201 - even though you are quoting what a participant may have said, I believe it is unprofessional and unnecessary to include the detail about 'having sex with your mother'. This is vulgar and offensive. You could just as easily write 'there would be no better way to die than having sex...no other than that I would prefer to be in my sleep' The message is the same, but the vulgarity removed
Line 207 privilege
Line 209 suggest you do not start the sentence with Whereas. I am also unsure how this paragraph, which incorporates one person's quote fits with the theme of 'Desirable Death'. The participant is talking about not fearing death and resolving their existence on earth. How does this relate to desirable death?
Line 223 to
Line 230 taken
Line 269-270 not knowing and not-knowing
Line 292 would be misled by a health care provider
ANALYSIS
Who did the transcription?
DISCUSSION
Line 345 understand
Line 398 suggest remove 'significantly'
line 399 remove 'as'. Same line you have 'I think, I think' - Even if this was said twice by the participant, i would suggest removing the duplication as it makes the reader do a 'double take'
Line 402 experience
Overall, I would say that the manuscript is significantly improved, but it is a shame that the authors did not proof read this version before submitting, as the majority of these errors would have been picked up. There is still some further clarification and re-ordering of data needed. I look forward to seeing this paper published
Author Response
Thank you so much for your willingness to review the manuscript again. We hope we have sufficiently addressed your concerns. The matrix we have attached includes the edits/responses for both you and Reviewer 1.
Best,
Andrea and Jessica

Round 3
Reviewer 2 Report
Thank you for your revision. This manuscript is much improved.